# Dietary Tributyrin Improves Growth Performance, Meat Quality, Muscle Oxidative Status, and Gut Microbiota in Taihe Silky Fowls under Cyclic Heat Stress

**DOI:** 10.3390/ani14203041

**Published:** 2024-10-21

**Authors:** Chuanbin Chen, Mingren Qu, Guanhong Li, Gen Wan, Ping Liu, Salma Mbarouk Omar, Wenliang Mei, Ziyu Hu, Qian Zhou, Lanjiao Xu

**Affiliations:** Jiangxi Province Key Laboratory of Animal Nutrition, Engineering Research Center of Feed Development, College of Animal Science and Technology, Jiangxi Agricultural University, Nanchang 330045, China; chenchuanb8297@163.com (C.C.); qumingren@jxau.edu.cn (M.Q.); liguanh@163.com (G.L.); wg8138@163.com (G.W.); pingliujx@163.com (P.L.); salmambarouk44@gmail.com (S.M.O.); 18942330983@163.com (W.M.); 13184620342@163.com (Z.H.); qianzhou182@163.com (Q.Z.)

**Keywords:** cyclic heat stress, Taihe silky fowls, tributyrin, growth performance, meat quality

## Abstract

**Simple Summary:**

High ambient temperature exposure-induced heat stress is one of the major problems in subtropical regions, severely affecting poultry health and production. This study was designed to evaluate the effects of dietary tributyrin supplementation on growth performance, meat quality, muscle oxidative status, and gut microbiota in Taihe silky fowls under cyclic heat stress (CHS) conditions. The results demonstrated that CHS remarkably reduced growth performance and impaired meat quality, while dietary tributyrin supplementation partially alleviated the adverse effects of CHS on these parameters. Additionally, tributyrin supplementation improved meat quality by enhancing muscle antioxidant capacity, which is believed to be associated with activating the Nrf2 signaling pathway. These findings are useful for the development of anti-heat stress feed additives in Taihe silky fowls.

**Abstract:**

Heat stress adversely affects poultry production and meat quality, leading to economic losses. This study aimed to investigate the effects of adding tributyrin on growth performance, meat quality, muscle oxidative status, and gut microbiota of Taihe silky fowls under cyclic heat stress (CHS) conditions. In this study, 120-day-old Taihe silky fowls (male) were randomly divided into six dietary treatments. These treatments included a normal control treatment (NC, fed a basal diet), a heat stress control treatment (HS, fed a basal diet), and HS control treatments supplemented with 0.04%, 0.08%, 0.16%, and 0.32% tributyrin, respectively. The NC treatment group was kept at 24 ± 1 °C, while the HS treatment birds were exposed to 34 ± 1 °C for 8 h/d for 4 weeks. Results showed that CHS decreased growth performance and compromised the meat quality of broilers (*p* < 0.05). However, tributyrin supplementation improved ADG and FCR in broilers exposed to CHS (*p* < 0.05). Additionally, tributyrin supplementation resulted in increased shear force value and GSH-Px activity, as well as a decrease in drip loss, ether extract content, and MDA content of the breast muscle in broilers under CHS (*p* < 0.05). Furthermore, tributyrin supplementation up-regulated the mRNA expressions of *Nrf2*, *NQO1*, *HO-1*, *SOD*, and *GSH-Px* of the breast muscle in broilers exposed to CHS (*p* < 0.05). Based on these positive effects, the study delved deeper to investigate the impact of 0.16% tributyrin supplementation (HS + 0.16%T) on the cecum microbiota. The HS + 0.16%T treatment showed an increase in the relative abundance of *Rikenellaceae_RC9_gut_group* (*p* < 0.05) and a trend towards an increase in *Lactobacillus* (*p* = 0.096) compared to the HS treatment. The results indicate that supplementation successfully improved the growth performance and meat quality of Taihe silky fowls. Furthermore, tributyrin supplementation, particularly at levels of 0.16%, improved meat quality by enhancing muscle antioxidant capacity, which is believed to be associated with activation of the Nrf2 signaling pathway.

## 1. Introduction

With global warming, high density, and the intensive poultry industry, the negative impact of heat-stressed broilers caused by high temperatures has attracted widespread attention worldwide, especially in tropical and subtropical regions [1]. Heat stress significantly affects the health and production of broilers and alters the composition of gut microbiota [2]. Furthermore, heat stress not only impairs the meat quality of broilers but also induces oxidative damage in their muscles [3]. In recent decades, there has been a growing emphasis on broiler meat quality, especially for Chinese consumers with improved living standards. As a result, indigenous slow-growing broiler chickens in China are increasingly preferred by consumers due to their better meat quality.

The Taihe silky fowl (*Gallus domestic us*) is a well-known indigenous Chinese slow-growing chicken breed. It has black skin, bones, and meat compared to other chickens [4]. One of its notable features is its richness in body melanin, which has been highly valued in China for over 2200 years as a curative food [5]. Recently, large-scale farming industries of Taihe silky fowls have started to take shape in China, and their consumption has been rapidly increasing [6]. The high environmental temperature in subtropical regions often causes heat stress in Taihe silky fowls, which may affect their health status, performance growth, and meat quality. To mitigate this, adjusting their diet, such as supplementing with feed additives, has been an effective strategy [7,8]. Butyrate has attracted widespread attention because of its nutritional and physiological functions [9,10].

Butyrate is a short-chain fatty acid found in the intestines of animals and which is derived from the fermentation of non-starch poly-saccharides [11]. It is quickly absorbed into the epithelial cells through the action of butyrate transporters [12] and provides energy for gut epithelial cells [13]. However, the unpleasant odor and the rapid metabolism of butyrate have limited its application in the livestock and poultry industry [14]. To overcome these drawbacks, there is increasing interest in developing butyric acid derivatives. Tributyrin is a derivative of butyrate, a natural product found in tallow. It is a valid alternative to butyrate, as one molecule of tributyrin releases three molecules of butyrate directly in the small intestine, and thus butyrate is rapidly adsorbed [15]. Studies have shown that supplementing tributyrin has positive effects on growth performance and gut health in animals [16,17,18]. Tributyrin has the ability to convert pepsinogen into pepsin, thereby increasing the activity of proteolytic enzymes and enhancing protein digestion [19]. Li et al. [20] also demonstrated that tributyrin alleviates the detrimental effects of lipopolysaccharide-challenged broilers by improving growth performance and anti-oxidant capacity. Muscle meat quality is related to anti-oxidants [21]. It has been reported that heat stress results in reducing the activity of anti-oxidant enzymes in the breast muscle of broilers [22]. Therefore, the negative effects of heat stress on meat quality may be related to changes in anti-oxidant capacity. To our knowledge, the effects of different levels of tributyrin on the growth performance and meat quality of Taihe silky fowls under heat stress have not been reported so far.

Therefore, this study aimed to investigate the effects of different dietary levels of tributyrin on the growth performance, nutrient digestibility, meat quality, and muscle oxidative status of Taihe silky fowls under CHS. Moreover, the effects of tributyrin on gut microbiota were evaluated.

## 2. Materials and Methods

### 2.1. Animal Ethics

The experimental procedures and protocols employed in this study were subjected to rigorous review and approval by the Committee for the Care and Use of Experimental Animals at Jiangxi Agricultural University (JXAULL-20210913).

### 2.2. Diet, Experiment Design, and Animals

Four hundred newly hatched Taihe silky fowls (male) were procured from a Taihe chicken seed farm and reared in battery brooders from 1 to 120 days of age. The broilers were fed commercial standard diets and subjected to standard husbandry practices. At 120 days of age, 300 healthy male broilers with similar body weights (915.29 ± 7.99 g) were randomly divided into six treatments, each of which was replicated as five cages with ten broilers per cage. These treatments included a normal control treatment (NC, fed a basal diet), a heat stress control treatment (HS, fed a basal diet), and HS control treatments supplemented with 0.04%, 0.08%, 0.16%, and 0.32% tributyrin, respectively, referred to as (HS + 0.04%T), (HS + 0.08%T), (HS + 0.16%T), and (HS + 0.32%T). The different levels of tributyrin addition refer to past studies [14,20]. The ambient temperature for the NC treatment group was maintained at 24 ± 1 °C, while HS treatment birds were exposed to 34 ± 1 °C for 8 h (9:30 am to 5:30 pm) per day, with the temperature reverting to 24 ± 1 °C for the remaining 16 h each day. The relative humidity for all treatments was maintained at 55 ± 5%. The birds in the NC and HS treatments were housed in different facilities, and the temperature and relative humidity of the NC and HS treatments were measured three times a day. Continuous artificial light illuminated the interior space for the whole experiment period. The chicken houses were equipped with environmental control equipment, and the size of the cage was 70 (length) × 70 (width) × 40 (height) cm. The adaptation period was 7 days, and the experimental period was 28 days. The tributyrin used in the present experiment was purchased from Perstorp Waspik B.V. (Waspik, the Netherlands) and contained 51.1% butyrate. The basal diet (Table 1) was formulated according to the Chinese Feeding Standard of Chicken, China (NY/T 33-2004), and tributyrin was thoroughly mixed into the diet through stepwise mixing before feeding. All birds had ad libitum access to water. The ambient temperature and humidity were recorded daily during the 28-day experimental period and are shown in Figure 1.

### 2.3. Temperature and Humidity

During the trial, the average ambient temperature for the NC treatment and HS treatment groups was 24.59 ± 0.66 °C and 33.86 ± 0.31 °C, and the average relative humidity was 58.65 ± 4.11% and 58.18 ± 3.03%, respectively.

### 2.4. Sample Collection

During the last 3 days of the experiment, the total excreta collection method was used for the determination of nutrient retention. During the test, feed intake and excrements were recorded daily, and the excreta were collected. Approximately 100 g of excreta and feed samples were collected. The samples were dried for 48 h in a dryer at 65 °C, followed by 24 h of equilibration to atmospheric conditions. Following this, the samples were pulverized using a Willey mill, passed through a 40-mesh screen, and stored at −20 °C to analyze nutrient digestibility.

At the end of the experiment, after 12 h fasting, 5 birds per treatment (one bird per cage was randomly selected and tagged) had blood samples collected (*n* = 30) from the wing vein. Serum was obtained through centrifugation (3500 r/min, 10 min, 4 °C), and stored at −20 °C for further analysis. After collecting blood samples, the broilers were slaughtered by severing the jugular vein, respectively. The meat quality of the slaughtered broilers was determined. Each broiler’s entire left breast meat was promptly excised and stored at 4 °C for subsequent analysis. Additionally, right breast muscle and cecal content samples were collected and rapidly stored in liquid nitrogen for further analysis.

### 2.5. Growth Performance Determination

Body weight (BW) was recorded on a per-cage basis on day 1 and 28. Feed consumption was recorded on a per-cage basis every week to calculate average daily feed intake (ADFI) and feed conversion ratio (FCR).

### 2.6. Nutrient Digestibility Determination

The dry matter (DM), crude protein (CP), and ether extract (EE) contents of both feed and excreta samples were analyzed using the AOAC (2000) method [23]. DM content was determined by oven-drying, CP content was determined using an auto-Kjeldahl nitrogen analyzer (FOSS/KT8400), and EE content was determined by Soxhlet extraction. Subsequently, the apparent digestibility (AD) of DM, CP, and EE was calculated using the following equation.
$$\mathrm{AD}\;(\%)=\frac{NI-NE}{NI}\times 100\%$$

Nutrient intake (*NI*) and nutrient excretion (*NE*) were derived from diet and excreta.

### 2.7. Serum HSP 70 and CORT Levels Determination

Serum heat shock protein 70 (HSP 70) and corticosterone (CORT) levels were analyzed by Chicken HSP70 and CORT ELISA assay kit (Shanghai Enzyme-linked Biotechnology Co., Ltd., Shanghai, China).

### 2.8. Meat Quality Determination

At 45 min and 24 h post-slaughter, the pH value of breast muscle was measured using a portable pH meter (HI9125). At 45 min post-slaughter, the color parameters of the breast meat, including lightness (L*), redness (a*), and yellowness (b*), were measured using a colorimeter (CR-400). The drip loss and cooking loss of the breast meat were measured 24 h post-slaughter, following the methods described by Lu et al. [22]. Lastly, the shear force value of the breast muscle was measured using a digital meat tenderness meter (C-LM4).

### 2.9. Breast Muscle Chemical Composition Determination

Moisture, CP, and EE contents in the breast meat were determined according to the method of AOAC (2000) [23].

### 2.10. Oxidative Status in the Breast Muscle Determination

The breast meat samples were homogenized and centrifugated (3500 r/min, 10 min, 4 °C). The resulting 10% tissue homogenate supernatant was collected. Glutathione peroxidase (GSH-Px), superoxide dismutase (SOD) activities, as well as total antioxidant capacity (T-AOC) and malondialdehyde (MDA) content were determined using commercial kits (Nanjing Jiancheng Institute of Bioengineering, Nanjing, China), following the manufacturer’s instructions. The protein concentration of the 10% tissue homogenate supernatant was determined using the bicinchoninic acid (BCA) assay kit (Nanjing Jiancheng Institute of Bioengineering, Nanjing, China).

### 2.11. Real-Time PCR Analysis

Total RNA was extracted from the breast muscle samples using TransZol reagent (Beijing TransGen Biotech Co., Ltd., Beijing, China). Once the concentration and purity of the total RNA were measured, it was reverse-transcribed into cDNA using a cDNA Synthesis kit (TransGen Biotech, Beijing, China).

Real-time qPCR analysis was performed using the CFX Connect Real-Time PCR Detection System (Bio-Rad, Hercules, CA, USA). The cDNA served as the template for real-time qPCR. The qPCR reaction mixture consisted of 10 μL of PerfectStartTM Green qPCR SuperMix (TransGen Biotech, Beijing, China), 0.4 μL each of forward and reverse primers, 2 μL of cDNA, and 7.2 μL of nuclease-free water, resulting in a total reaction system of 20 μL. The PCR reaction procedure was guided as follows: 95 °C for 30 s, 42 cycles of 95 °C for 5 s, and 60 °C for 30 s. Specific primers listed in Table 2, were designed and synthesized by Shanghai Generay Biotechnology Co., Ltd. (Shanghai, China) in this study. The expression levels of target genes were normalized to β-actin using the 2^−ΔΔCt^ method.

### 2.12. DNA Extraction and Sequencing

Genomic DNA was extracted from the cecal content samples of the NC treatment, HS treatment, and HS + 0.16%T treatment using an E.Z.N.A. Stool DNA Kit (Omega Bio-tek, Inc., Norcross, GA, USA) following the manufacturer’s manual. The V3-V4 hypervariable regions of the 16S rRNA gene were amplified with the primer of 338F (5′-ACTCCTACGGGAGGCAGCAG-3′) and 806R (5′-GGACTACNNGGGTATCTAAT-3′). Sequencing libraries were generated using the NEB Next Ultra II DNA Library Prep Kit (New England Biolabs, Inc., Ipswich, MA, USA) following the manufacturer’s recommendations. The quality of the libraries was assessed by Nanodrop 2000 (ThermoFisher Scientific, Inc., Waltham, MA, USA), Agilent 2100 Bioanalyzer (Agilent Technologies, Inc., Santa Clara, CA, USA), and ABI StepOnePlus Real-Time PCR System (Applied Biosystems, Inc., Waltham, MA, USA), successively. Deep sequencing was performed using the Illumina Miseq/Novaseq 6000 platform (Illumina Inc., San Diego, CA, USA).

Quantitative Insights into Microbial Ecology (QIIME) software version 1.8.0 [24] was utilized to process raw data for sequencing. The raw data were filtered and spliced using Pear software version 0.9.6 [25]. Any sequences with ambiguous bases or shorter than 120 bp were removed. After splicing, Vsearch software v2.7.1 [26] was used to remove sequences less than 230 bp in length and removed the chimeric sequence by the UCHIME [27] method according to the Gold Database. Sequences with a similarity > 97% were clustered into operational taxonomic units (OTUs) using UPARSE [28]. The BLAST [29] tool was used to classify all OTU representative sequences into different taxonomic groups against the Silva138 Database. Alpha diversity and beta diversity analysis were further conducted based on the OUT information from each sample.

### 2.13. Statistical Analyses

The growth performance, nutrient digestibility, blood biochemical parameters, meat quality, muscle antioxidant capacity, and gene expression data were analyzed using the SPSS statistics software package 25.0 (IBM software, Chicago, IL, USA). An independent sample T-test was used for comparing the NC treatment and the HS treatment. * *p* < 0.05 indicates a significant difference. A one-way analysis of variance (ANOVA) and Tukey’s test were used for comparing the different levels of tributyrin. Variability in data was expressed as standard error of means (SEM); *p* < 0.05 was considered significant, while 0.05 ≤ *p* < 0.10 was considered a trend.

Relative abundances of OTUs in each sample’s data were firstly conducted through a normal distribution test using the SPSS procedure, followed by differential analysis between the NC treatment and the HS treatment, as well as between the HS treatment and the HS + 0.16%T treatment through an independent sample T-test using SPSS statistics software package 25.0 (IBM software, Chicago, IL, USA). Alpha diversity and beta diversity for our samples were calculated using QIIME (version 1.88.0) and displayed with R software (version 3.6.0, R Core Team, Vienna, Austria). Using Bray Curtis algorithms in R software, principal coordinate analysis (PCoA) was employed to determine the discrepancies between samples.

## 3. Results

### 3.1. Subsection

#### 3.1.1. Growth Performance

In Table 3, it is evident that broilers in the HS treatment had significantly lower FBW, ADFI, ADG, and FCR compared with the NC treatment (*p* < 0.05). Additionally, supplementation with 0.16% and 0.32% tributyrin significantly enhanced FBW, ADG, and FCR of broilers during CHS (*p* < 0.05). However, tributyrin supplementation did not have an impact on the ADFI of broilers under CHS (*p* > 0.05).

#### 3.1.2. Nutrient Digestibility

In Table 4, it is evident that compared to the NC treatment, the digestibility of DM and CP in broilers significantly decreased in the HS treatment (*p* < 0.05). Additionally, supplementation with 0.08%, 0.16% and 0.32% tributyrin increased CP digestibility in broilers under CHS (*p* < 0.05).

#### 3.1.3. Serum HSP70 and CORT Levels

In Table 5, compared to the NC treatment, serum HSP70 and CORT levels significantly increased in the HS treatment (*p* < 0.05). Additionally, supplementation with 0.16% and 0.32% tributyrin increased serum HSP70 and CORT levels in broilers under CHS (*p* < 0.05).

#### 3.1.4. Meat Quality and Breast Muscle Chemical Composition

In Table 6, HS significantly increased drip loss and cooking loss of breast meat and significantly decreased pH_45 min_ and shear force value of breast meat compared with the NC treatment (*p* < 0.05). Additionally, supplementation with 0.08%, 0.16%, and 0.32% tributyrin significantly reduced drip loss (*p* < 0.05) and increased the shear force value of breast muscle in broilers under CHS (*p* < 0.05).

The chemical composition results in the breast muscle are shown in Table 7. Compared with the NC treatment, HS significantly increased the EE content and decreased the CP content (*p* < 0.05) of breast meat. Moreover, supplementation with 0.16% and 0.32% tributyrin decreased the EE content (*p* < 0.05) of breast muscle in broilers under CHS.

#### 3.1.5. Oxidative Status in the Breast Muscle

In Table 8, HS significantly decreased the activities of SOD and GSH-Px and significantly increased the MDA content of breast muscle compared to the NC treatment (*p* < 0.05). Additionally, supplementation with 0.16% and 0.32% tributyrin decreased the MDA content and increased the GSH-Px activity of breast muscle in broilers under CHS (*p* < 0.05).

#### 3.1.6. Gene Expression of Antioxidant Enzymes in the Breast Muscle

In Figure 2, it was observed that HS significantly down-regulated the mRNA expressions of *Nrf2*, *NQO1*, *HO-1*, *SOD*, and *GSH-Px* of breast muscle compared to the NC treatment (*p* < 0.05). Moreover, supplementation with 0.16% and 0.32% tributyrin up-regulated mRNA expressions of *Nrf2*, *NQO1*, *HO-1*, *SOD*, and *GSH-Px* (*p* < 0.05) of breast muscle in broilers under CHS.

#### 3.1.7. Cecum Microbiota

The filtering method found 30 phyla and over 260 genera, which were used to assess biodiversity.

Alpha diversity (Figure 3), reflecting the internal complexity of each treatment, was assessed. There were no significant differences in diversity indices (Chao1, Observed species, Shannon, and Simpson) between the NC treatment and the HS treatment, nor between the HS treatment and the HS + 0.16%T treatment (*p* > 0.05).

At the phylum level (Figure 4), caeca bacteria were dominated by *Bacteroidota*, followed by *Firmicutes* and *Proteobacteria.* Concretely speaking, compared with the NC treatment, there was a trend towards an increase in the relative abundance of *Desulfobacterota* (*p* = 0.090) and a trend towards a decrease in the relative abundance of *Fusobacteriota* (*p* = 0.077) were observed in the HS treatment.

At the genus level (Figure 5). Broilers in the HS treatment had higher relative abundances of *Parabacteroides*, *Parabacteroides*, and *Treponem* (*p* < 0.05), and lower abundances of *Rikenellaceae_RC9_gut_group* and *Ruminococcus_torques_group*, *Fusobacteriu* (*p* < 0.05) compared to the NC treatment. In addition, there was a trend towards a decrease in the relative abundance of *Fusobacterium* (*p* = 0.077) and *Lactobacillus* (*p* = 0.052) was observed in the HS treatment compared to the NC treatment. Comparatively, the HS + 0.16%T treatment showed an increase in the relative abundance of *Rikenellaceae_RC9_gut_group* (*p* < 0.05) and a trend towards an increase in *Lactobacillus* (*p* = 0.096) compared to the HS treatment.

## 4. Discussion

### 4.1. Successful Construction of Heat Stress Models

HSP70 is a key stress response protein found in various cells and tissues. When poultry experiences high-temperature stress, the expression of HSP70 increases along with the protein content [30]. CORT is secreted by the adrenal cortex under the stimulation of the hypothalamic–pituitary–adrenal (HPA) axis and serves as a useful indicator for monitoring the stress status of poultry [31]. Multiple studies have indicated that heat stress leads to an increase in serum CORT levels [32,33]. In the present study, cyclic high temperatures (34 ± 1 °C for 8 h/day over 28 days) led to increased levels of serum HSP70 and CORT in broilers compared with non-HS treatment, confirming the successful induction of heat stress in broilers.

### 4.2. Growth Performance, Nutrient Digestibility, and Cecum Microbiota

The high temperature has a significant impact on the production of broilers [1]. Our data indicated that CHS significantly decreased the FBW, ADFI, ADG, and FCR of broilers, which is consistent with the results of Liu et al. [34], who showed a decrease in ADFI, ADG, and FCR of broilers under CHS. During heat stress, heat production is associated with lower nutrient utilization efficiency because of decreased FI [35]. The gut microbiota is a key factor influencing nutrient intake in the intestine. We found that CHS significantly increased the relative abundance of Treponema in the cecum of broilers. Treponema belongs to the Spirochaetes phylum, and their abundance increases with heat stress [36]. It has been reported that Treponema is positively correlated with nutrient digestibility in pigs [37]. As expected, broilers exposed to CHS exhibited lower digestibility of dry matter in our study.

Butyrate is a volatile fatty acid that plays a vital role in maintaining intestinal mucosal integrity, reducing inflammation, acting as an antioxidant, and improving the growth performance of broilers [9,38,39]. Tributyrin entering the intestine can liberate three molecules of butyric acid that are rapidly absorbed by the intestinal epithelial cells [40]. A digestion experiment in vitro revealed that tributyrin only released a small amount of butyrate (<5%) in the stomach phase and approximately 75% of the total butyrate in the small intestine [41], which showed that tributyrin had a higher utilization efficiency in the intestine. The present study found that adding tributyrin to the diet can positively influence the growth performance of broilers experiencing CHS, showing an increased ADG and FCR. Similarly, as derivatives of butyrate, supplementing the diet with sodium butyrate alleviated heat stress and increased both ADFI and ADG in broilers [42]. The improved growth performance in broilers could be attributed to enhanced intestinal morphology and increased nutrient digestibility resulting from the tributyrin addition to the diets. Tributyrin has the ability to convert pepsinogen into pepsin, thereby increasing the activity of proteolytic enzymes and enhancing protein digestion [19]. Our data also demonstrated that tributyrin supplementation increased CP digestibility in broilers under CHS. Additionally, another study reported that sodium butyrate dietary supplementation improved the intra-luminal digestibility of proteins [43].

The composition of intestinal microflora plays an important role in gut health [44]. Heat stress can disturb the balance of the intestinal microflora, which allows opportunistic pathogens to multiply and cause intestinal disorders [35]. So far, few data are available or have been published regarding the effect of tributyrin on the intestinal microflora of broilers under CHS. Our study found that supplementing with tributyrin increased the relative abundance of Rikenellaceae_RC9_gut_group and showed a trend towards an increase in Lactobacillus in broiler cecum under CHS. Lactobacillus is an important probiotic, and its increase is considered beneficial for gut health [45]. Lactobacillus can assist broilers hosts in obtaining nutritional support, reducing pH levels in the intestines, maintaining the mucosal barrier function, and protecting against damage from foreign substances and pathogens [46,47]. A study reported that a higher abundance of cecal Lactobacillus positively regulated feed efficiency [48]. In addition, Rikenellaceae_RC9_gut_group produces propionate and butyrate from succinate or lactate as substrates [49]. Functionally, Rikenellaceae_RC9_gut_group regulates lipid metabolism, glucose metabolism homeostasis [50], and alleviates oxidative stress [51]. These results suggest that tributyrin can modulate gut microbiota imbalance, increase energy provision for physiological activities, improve the growth performance of broilers.

### 4.3. Meat Quality

Drip loss and cooking loss serve as crucial measures in assessing the water-holding capability of muscle [3]. Post-slaughter pH is also a crucial factor in meat quality [52]. We observed that CHS increased drip loss and cooking loss, and decreased pH_45 min_ of breast muscle in broilers. Similarly, a study by Lu et al. [22] showed that CHS (32 °C for 14 days) decreased pH_45 min_, pH_24 h_, and shear force value while increasing drip loss of breast muscle in broilers. Meanwhile, Wen et al. [53] found that CHS (34 °C for 21 days) increased drip loss of breast meat in broilers. These changes may be attributed to heat stress accelerating glycolysis, leading to more pyruvate in muscle to be converted into lactic acid and resulting in decreased post-slaughter pH [54]. Furthermore, a low post-slaughter pH can degrade muscle proteins, subsequently increasing muscle drip loss and cooking loss [22], which aligns with the findings of this study.

High temperature has a negative impact on the chemical composition of muscles, and low post-slaughter pH combined with poorer water-holding capacity of muscle in broilers can lead to the loss of liquid, soluble nutrients, and flavor, ultimately increasing the risk of pale, soft, exudative (PSE) meat [55]. Zhang et al. [56] reported lower CP content of broiler breast muscle under CHS, which is consistent with our results. A study reported that CHS increases the fat content of muscle as well as the mRNA expression levels of the key regulatory genes ACC and FAS in fatty acid synthesis [22]. This indicates that CHS increases the synthesis and deposition of fat of breast muscle in broilers. Chen et al. [3] showed that higher EE content of breast muscle occurs in broilers under CHS, which aligns with our findings. Additionally, a study has revealed a negative correlation between muscle fat content and their shear force and water-holding capacity [57], which may explain the decrease in the shear force value of broiler breast muscle under CHS in this study. We found that tributyrin supplementation decreased drip loss and EE content and increased shear value in the breast meat of broilers under CHS. Lan et al. [42] reported that sodium butyrate supplementation decreased muscular drip loss of broilers under hot climatic conditions, indicating that tributyrin could mitigate the negative impact on the meat quality of broilers under CHS. This suggests that the enhanced oxidative status of breast muscle is associated with the supplementation of dietary tributyrin.

### 4.4. Muscle Oxidative Status

The oxidative status of muscle is also crucial for meat quality [3]. It is widely reported that heat stress has negative effects on muscle oxidative status [21,22,42]. Our data again indicated that CHS compromised the antioxidant capacity in the breast meat of broilers. This is evidenced by increased MDA content and decreased activities of SOD and GSH-Px in breast meat. Undoubtedly, heat stress induces the overproduction of free radicals [22], accelerating substrate oxidation [3] and causing damage to muscle cell membrane integrity and mitochondrial function [58], ultimately resulting in muscle damage caused by oxidative stress. The meat of Taihe silky fowls contains melanin when compared with other common chickens [6]. Natural melanin is considered as one of the most important components in Taihe silky fowls, which has a wide range of biochemical activities such as anti-oxidation and free radical-scavenging effects [5]. Liu et al. [4] also reported a significant increase in MDA and lower levels of antioxidant enzymes in black-boned chickens exposed to high temperatures. This may be due to the decrease of melanin synthesis in Taihe black-bone chickens due to heat stress. Nrf2 is a crucial regulator that augments cellular defense against oxidative damage. Under oxidative stress conditions, Nrf2 dissociates from Keap-1, enters the nucleus, and interacts with antioxidant response elements (ARE) to activate the transcription of downstream genes [59]. Genes containing ARE sequences, including *NQO1*, *HO-1*, *GSH*-related genes, *SOD*, and *CAT*, can all be activated and transcribed by Nrf2 to exert antioxidant functions [60]. Our data observed that CHS down-regulated the mRNA expressions of *Nrf2*, *NQO1*, *HO-1*, *SOD*, and *GSH-Px* of breast meat in broilers. Chen et al. [3] also showed that heat stress down-regulated the mRNA expressions of *Nrf2*, *NQO1*, *Cu/ZnSOD*, and *GSH-Px* of breast meat in broilers. Therefore, these results directly explain the muscle damage of broilers under CHS.

Our study found that tributyrin supplementation significantly reduced MDA content and increased GSH-Px activity of breast muscle in broilers under CHS. Lan et al. [42] reported that adding sodium butyrate decreased MDA content and increased the activities of SOD and GSH-Px of breast meat in broilers under hot climatic conditions. Wang et al. [61] reported that, under thermo-neutral temperatures, adding tributyrin significantly increased T-AOC and decreased MDA levels in the ovaries of broilers. This implies that tributyrin supplementation may effectively alleviate muscle oxidative damage of broilers under CHS. In addition, activating Nrf2 and its downstream gene expression would reduce the generation of free radicals and decrease lipid peroxidation, which may further improve antioxidant status [62]. In the present study, tributyrin supplementation significantly up-regulated the mRNA expression of *Nrf2*, *NQO1*, *HO-1*, *SOD*, and *GSH-Px* of breast muscle in broilers under CHS, suggesting that tributyrin would improve the antioxidant capacity of breast meat, possibly, at least partially, via activating the Nrf2 signaling pathway.

However, in our study, dietary tributyrin supplementation had no significant effects on pH, color parameters, or cooking loss of breast muscle. This aspect could be explained based on a relatively insufficient number of observations as well as the limited criteria used to evaluate meat quality. These limitations of our study are acknowledged, thus increasing the sample size of the experiment, and investigating other criteria related to meat quality, such as glycogen, lactic acid, lactic dehydrogenase, citrate synthase, pyruvate kinase, etc., are essential in future studies. Furthermore, the in vivo experiment demonstrated tributyrin’s ability to alleviate the oxidative damage of broiler breast muscles caused by heat stress, possibly via Nrf2 signaling pathway activation. Still, Nrf2 involvement had not been verified and building an in vitro model of broiler’s skeletal muscle cells would serve this purpose. In addition, the effects of different tributyrin levels (0.04%, 0.08%, 0.16%, and 0.32%) of supplementation on broilers under thermoneutral temperature conditions were not considered. Still, Gu et al. [14] reported that a 1% tributyrin-supplemented diet did not cause tissue damage under thermoneutral temperature conditions. Hou [63] et al. also reported that supplementation with 1% tributyrin can effectively improve the growth performance, serum antioxidant capacity, intestinal morphology, and cecal microbiota composition of yellow-feathered broilers under thermoneutral temperature conditions.

## 5. Conclusions

In conclusion, tributyrin dietary supplementation modulated gut microbiota imbalance, enhanced nutrient digestibility, and improved growth performance and meat quality in Taihe silky fowls under CHS. Furthermore, supplementation with 0.16% and 0.32% tributyrin was effective in improving meat quality by enhancing muscle antioxidant capacity, which is believed to be associated with activating the Nrf2 signaling pathway. Considering the production cost comprehensively, the optimum supplemental level of tributyrin was 0.16% under the conditions of this study.

## Figures and Tables

**Figure 1 animals-14-03041-f001:**
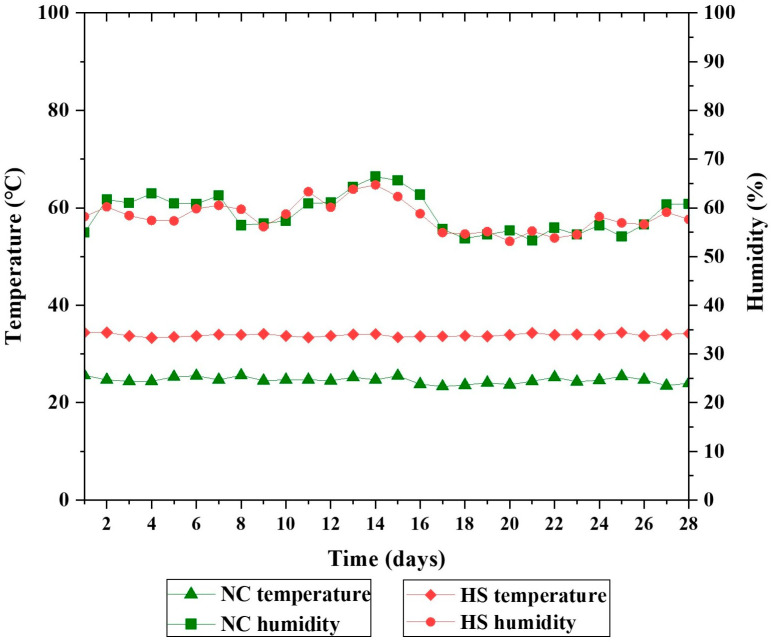
Average ambient temperature and average relative humidity registered for the NC treatment group and the HS treatment group during their respective 28-day periods. NC = normal control treatment; HS = heat stress treatment.

**Figure 2 animals-14-03041-f002:**
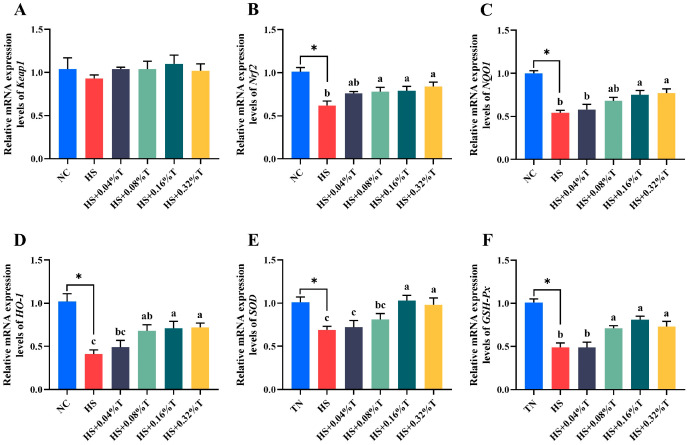
Effects of dietary tributyrin supplementation on the expression of antioxidant-related genes (**A**) Keap1, (**B**) Nrf2, (**C**) NQO1, (**D**) HO-1, (**E**) SOD, and (**F**) GSH-Px, respectively, in the breast muscle of Taihe silky fowls under CHS. NC = normal control treatment; HS = heat stress treatment; HS + 0.04%T = HS treatment supplemented with 0.04% tributyrin; HS + 0.08%T = HS treatment supplemented with 0.08% tributyrin; HS + 0.16%T = HS treatment supplemented with 0.16% tributyrin; HS + 0.32%T = HS treatment supplemented with 0.32% tributyrin. Results are mean value ± standard error. An “*” indicates a significant difference (*p* < 0.05) for the NC treatment vs. the HS treatment. Mean values with different small letters denote significant differences (*p* < 0.05) among HS, HS + 0.04% T, HS + 0.08%, HS + 0.16% T, and HS + 0.32% T treatments.

**Figure 3 animals-14-03041-f003:**
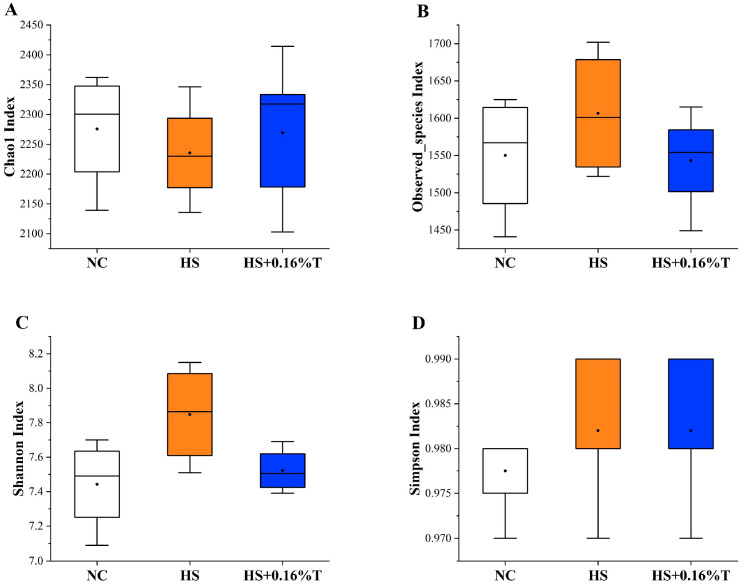
Effects of dietary tributyrin supplementation on the alpha diversity indices representing (**A**) Chao1 index, (**B**) Observed species index, (**C**) Shannon index, (**D**) Simpson index in the cecum of Taihe silky fowls under CHS. NC = normal control treatment; HS = heat stress treatment; HS + 0.16%T = HS treatment supplemented with 0.16% tributyrin.

**Figure 4 animals-14-03041-f004:**
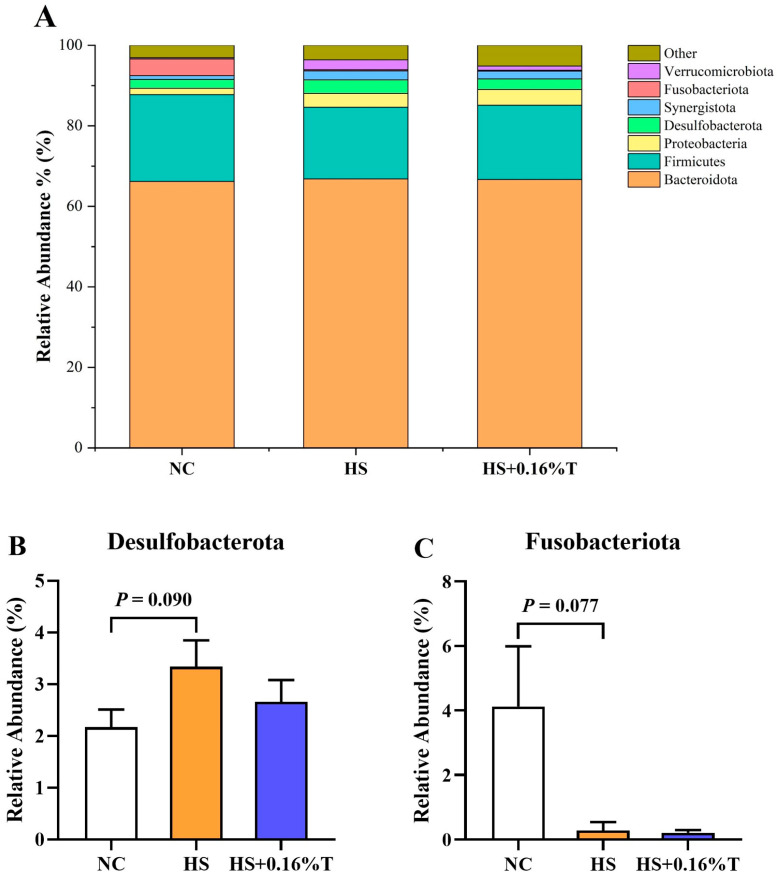
Effects of dietary tributyrin supplementation on the (**A**) relative abundance of phylum and significant abundance microbiota, namely (**B**) Desulfobacterota and (**C**) Fusobacteriota in the cecum of Taihe silky fowls under CHS. NC = normal control treatment; HS = heat stress treatment; HS + 0.16%T = HS treatment supplemented with 0.16% tributyrin.

**Figure 5 animals-14-03041-f005:**
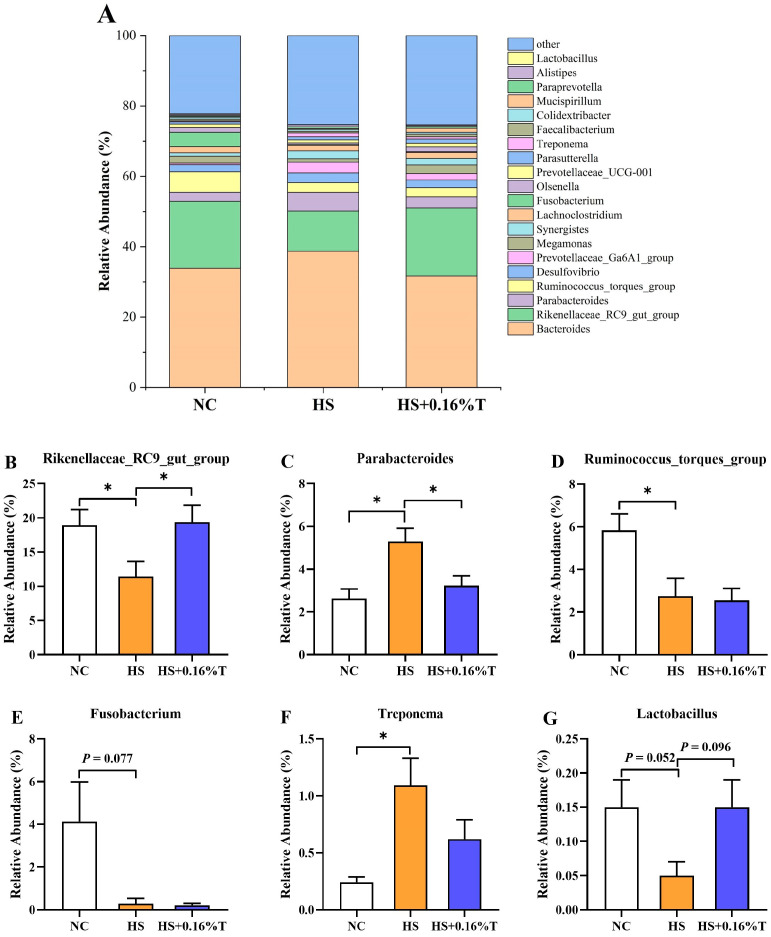
Effects of dietary tributyrin supplementation on the (**A**) relative abundance of the top twenty most abundant genera and significant abundance of microbiota, namely (**B**) Rikenellaceae_RC9_gut_group, (**C**) Parabacteroides, (**D**) Ruminococcus_torques_group, (**E**) Fusobacterium, (**F**) Treponema, (**G**) Lactobacillus, respectively, in the cecum of Taihe silky fowls under CHS. NC = normal control treatment; HS = heat stress treatment; HS + 0.16%T = HS treatment supplemented with 0.16% tributyrin. An “*” indicates a significant difference (*p* < 0.05) for the NC treatment vs. the HS treatment and the HS treatment vs. the HS + 0.16%T treatment.

**Table 1 animals-14-03041-t001:** Composition and nutrient levels of the basal diet (air-dry basis).

Ingredients (%)		Calculated Nutrient Levels	
Corn	67.60	Metabolizable energy (MJ/kg)	12.97
Soybean meal	22.00	Crude protein (%)	16.19
Fish meal	2.00	Calcium (%)	0.81
Soybean oil	4.00	Available phosphorus (%)	0.38
Dicalcium phosphate	1.00	Threonine (%)	0.69
Limestone	1.10	Lysine (%)	0.86
DL-Methionine	0.09	Methionine (%)	0.36
Threonine	0.05	Methionine + cystine (%)	0.65
Salt	0.30		
Zeolite powder	1.36		
Premix *	0.50		
Total	100.00		

* The premix provided per kilogram of diet: 11,250 IU of vitamin A, 3750 IU of vitamin D3, 30 IU of vitamin E, 1.75 mg of vitamin K3, 2.75 mg of thiamin, 9.30 mg of riboflavin, 18.75 mg of calcium pantothenate, 37.50 mg of nicotinamide, 3.75 mg of vitamin B6, 0.15 mg of biotin, 1.80 mg of folic acid, 0.028 mg of vitamin B12, 720 mg of choline chloride, 80 mg of Fe (from ferrous sulfate), 8 mg of Cu (from cupric chloride), 80 mg of Mn (from manganous sulfate), 60 mg of Zn (from zinc sulfate), 0.35 mg of I (from calcium iodate), and 0.15 mg of Se (from sodium selenite).

**Table 2 animals-14-03041-t002:** Information of target genes and primers.

Target Genes	Accession No.	Primer Sequences (5′ to 3′ Direction)
Keap1	MN416132.1	F: ATGTACCAGATCGACAGCGT R: AACTCCTCCTGCTTGGAGAC
Nrf2	MN416129.1	F: GCCACCCTAAAGCTCCATTC R: ACTGAACTGCTCCTTCGACA
NQO1	NM_001277621.1	F: CCTCTACGCCATAGGGTTCA R: TGCAGTGGGAACTGGAAGAT
HO-1	NM_205344.1	F: GAAAGCTGCCCTGGAGAAAG R: CCCAGACAGGTCTCCCAAAT
SOD	NM_205064.1	F: ATGTGACTGCAAAGGGAGGA R: AGCTAAACGAGGTCCAGCAT
GSH-Px	NM_001277853.2	F: ACGGCTTCAAACCCAACTTC R: CTCTCTCAGGAAGGCGAACA
β-actin	NM_205518.1	F: ATCAGCAAGCAGGAGTACGA R: AAAGCCATGCCAATCTCGTC

**Table 3 animals-14-03041-t003:** Effects of dietary tributyrin supplementation on growth performance of Taihe silky fowls under cyclic heat stress.

Items	NC	HS + T					NC vs. HS	HS + T	
		0%	0.04%	0.08%	0.16%	0.32%	*p*-Value	SEM	*p*-Value
IBW	916.40	913.50	916.00	916.50	917.60	911.60	0.592	1.77	0.841
FBW	1173.33	1052.78 ^b^	1073.33 ^ab^	1079.86 ^ab^	1097.22 ^a^	1099.11 ^a^	<0.001	5.54	0.039
ADFI (g/d)	50.78	43.33	43.06	43.72	44.33	44.69	0.011	0.52	0.874
ADG (g/d)	9.35	4.97 ^b^	5.62 ^ab^	5.84 ^ab^	6.42 ^a^	6.53 ^a^	<0.001	0.19	0.046
FCR	5.44	8.76 ^a^	7.81 ^ab^	7.55 ^b^	6.92 ^b^	6.96 ^b^	<0.001	0.20	0.012

NC = normal control treatment; HS = heat stress treatment; HS + T = HS treatment supplemented (0%, 0.04%, 0.08%, 0.16%, 0.32%) with tributyrin; SEM, standard error of means; NC vs. HS, NC treatment vs. HS control (0% tributyrin) treatment. IBW = initial body weight; FBW = final body weight; ADFI = average daily feed intake; ADG = average daily gain; FCR = feed conversion ratio. Mean values with different letters were significantly different (*p* < 0.05).

**Table 4 animals-14-03041-t004:** Effects of dietary tributyrin supplementation on nutrient digestibility of Taihe silky fowls under cyclic heat stress.

Items	NC	HS + T					NC vs. HS	HS + T	
		0	0.04	0.08	0.16	0.32	*p*-Value	SEM	*p*-Value
Dry matter (%)	73.57	71.60	72.50	72.12	72.50	72.24	0.038	0.30	0.876
Crude protein (%)	46.89	41.54 ^b^	42.20 ^ab^	44.28 ^a^	44.77 ^a^	44.68 ^a^	<0.001	0.44	0.035
Ether extract (%)	82.99	82.05	81.57	81.78	82.51	82.47	0.252	0.28	0.808

NC = normal control treatment; HS = heat stress treatment; HS + T = HS treatment supplemented (0%, 0.04%, 0.08%, 0.16%, 0.32%) with tributyrin; SEM, standard error of means; NC vs. HS, NC treatment vs. HS control (0% tributyrin) treatment. Mean values with different letters were significantly different (*p* < 0.05).

**Table 5 animals-14-03041-t005:** Effects of dietary tributyrin supplementation on serum HSP 70 and CORT levels of Taihe silky fowls under cyclic heat stress.

Items	NC	HS + T					NC vs. HS	HS + T	
		0	0.04	0.08	0.16	0.32	*p*-Value	SEM	*p*-Value
HSP 70 (pg/mL)	150.36	234.42 ^a^	212.28 ^ab^	206.60 ^ab^	172.58 ^b^	178.04 ^b^	0.001	6.99	0.017
CORT (ng/mL)	3.63	4.76 ^a^	4.59 ^a^	4.46 ^ab^	3.66 ^b^	3.63 ^b^	0.005	0.15	0.026

NC = normal control treatment; HS = heat stress treatment; HS + T = HS treatment supplemented (0%, 0.04%, 0.08%, 0.16%, 0.32%) with tributyrin; SEM, standard error of means; NC vs. HS, NC treatment vs. HS control (0% tributyrin) treatment. HSP 70 = heat shock protein 70; CORT = corticosterone. Mean values with different letters were significantly different (*p* < 0.05).

**Table 6 animals-14-03041-t006:** Effects of dietary tributyrin supplementation on meat quality of breast meat in Taihe silky fowls under cyclic heat stress.

Items	NC	HS + T					NC vs. HS	HS + T	
		0%	0.04%	0.08%	0.16%	0.32%	*p*-Value	SEM	*p*-Value
pH_45 min_	6.46	6.37	6.40	6.43	6.40	6.39	0.042	0.01	0.760
pH_24 h_	6.07	6.12	6.19	6.15	6.13	6.13	0.143	0.01	0.368
Lightness (L*)	43.64	43.42	43.96	43.08	43.20	43.39	0.806	0.36	0.974
Redness (a*)	1.66	1.40	1.48	1.47	1.46	1.48	0.367	0.10	0.999
Yellowness (b*)	4.05	3.31	3.23	3.42	3.77	3.81	0.369	0.21	0.912
Drip loss (%)	1.28	1.69 ^a^	1.54 ^ab^	1.49 ^b^	1.47 ^b^	1.43 ^b^	0.003	0.03	0.035
Cooking loss (%)	13.13	14.37	14.89	14.61	14.27	14.38	0.268	0.34	0.989
Shear force (N)	17.22	14.65 ^b^	15.15 ^ab^	15.80 ^a^	16.09 ^a^	15.88 ^a^	0.003	0.17	0.034

NC = normal control treatment; HS = heat stress treatment; HS + T = HS treatment supplemented (0%, 0.04%, 0.08%, 0.16%, 0.32%) with tributyrin; SEM, standard error of means; NC vs. HS, NC treatment vs. HS control (0% tributyrin) treatment. pH_45 min_ = pH at 45 min post-mortem; pH_24 h_ = pH at 24 h post-mortem. Mean values with different letters were significantly different (*p* < 0.05).

**Table 7 animals-14-03041-t007:** Effects of dietary tributyrin supplementation on chemical composition of breast meat in Taihe silky fowls under cyclic heat stress.

Items	NC	HS + T					NC vs. HS	HS + T	
		0%	0.04%	0.08%	0.16%	0.32%	*p*-Value	SEM	*p*-Value
Moisture (%)	69.30	69.72	70.28	70.18	70.10	69.63	0.487	0.15	0.594
Ether extract (%)	1.84	2.66 ^a^	2.46 ^ab^	2.43 ^ab^	2.12 ^b^	2.14 ^b^	0.029	0.07	0.047
Crude protein (%)	29.75	28.65	28.25	28.45	28.64	28.57	0.005	0.13	0.874

NC = normal control treatment; HS = heat stress treatment; HS + T = HS treatment supplemented (0%, 0.04%, 0.08%, 0.16%, 0.32%) with tributyrin; SEM, standard error of means; NC vs. HS, NC treatment vs. HS control (0% tributyrin) treatment. Mean values with different letters were significantly different (*p* < 0.05).

**Table 8 animals-14-03041-t008:** Effects of dietary tributyrin supplementation on antioxidant capacities of breast meat in Taihe silky fowls under cyclic heat stress.

Items	NC	HS + T					NC vs. HS	HS + T	
		0%	0.04%	0.08%	0.16%	0.32%	*p*-Value	SEM	*p*-Value
T-AOC (mmol/g protein)	0.106	0.101	0.105	0.104	0.105	0.104	0.516	0.003	0.995
SOD (U/mg protein)	34.05	25.48	25.50	27.21	27.64	27.55	0.004	0.59	0.628
GSH-Px (U/mg protein)	28.55	11.83 ^b^	15.61 ^ab^	19.17 ^ab^	24.50 ^a^	24.74 ^a^	<0.001	1.64	0.028
MDA (nmol/mg protein)	0.123	0.213 ^a^	0.187 ^ab^	0.191 ^ab^	0.177 ^b^	0.170 ^b^	<0.001	0.005	0.035

NC = normal control treatment; HS = heat stress treatment; HS + T = HS treatment supplemented (0%, 0.04%, 0.08%, 0.16%, 0.32%) with tributyrin; SEM, standard error of means; NC vs. HS, NC treatment vs. HS control (0% tributyrin) treatment. T-AOC = total antioxidant capacity; SOD = superoxide dismutase; GSH-Px = glutathione peroxidase; MDA = malondialdehyde. Mean values with different letters were significantly different (*p* < 0.05).

## Data Availability

The DNA sequences of this study were deposited in the National Center for Biotechnology Information (NCBI) Sequence Read Archive (SRA) repository under accession number PRJNA1168254.

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
