# Peer review of "Dietary Tributyrin Improves Growth Performance, Meat Quality, Muscle Oxidative Status, and Gut Microbiota in Taihe Silky Fowls under Cyclic Heat Stress"

_animals, 2024, doi:10.3390/ani14203041_

Round 1
Reviewer 1 Report
Comments and Suggestions for Authors
This study evaluated the effects of dietary supplementation tributyrin on growth performance, meat quality, muscle oxidative status, and gut microbiota in Taihe silky fowls under cyclic heat stress condition. The research was interesting and it can be accepted after revision.
1. Line 25. “24±1” should be “24±1℃”.
2.Line 101. How many broilers in each cage? Besides, we can not know the number of broilers in each assay.
3.Line 106. The authors stated that the ambient temperature for the NC treatment was maintained 24 ± 1℃. While in line 117, the authors stated that the average ambient temperature in the NC treatment was 24.59 ± 0.66℃.
4.The description of results part was not accurate and should be improved. For example (line 235-236), the authors stated that the inclusion of tributyrin significantly enhanced FBW, ADG, and FCR of broilers during CHS (p < 0.05). The fact was that not all the level of tributyrin supplementation had the effect.
5.Line 552. The authors stated that tributyrin supplementation, particularly at levels of 0.16% improved the meat quality by enhancing muscle antioxidant capacity. Why not 0.32%? How you come to this conclusion?
Author Response
Comments 1: Line 25. “24±1” should be “24±1℃”.
Response 1: Thank you for pointing this out. I agree with this comment. I have modified.
Comments 2: Line 101. How many broilers in each cage? Besides, we can not know the number of broilers in each assay.
Response 2: Thank you for pointing this out. I have rewritten this sentence for more detail.
Line 99-102: At 120 days of age, 300 healthy male Taihe silky fowls with similar body weights (915.29 ± 7.99 g) were randomly divided into six treatments, each of which was replicated five cages with ten broilers per cage.
Comments 3: Line 106. The authors stated that the ambient temperature for the NC treatment was maintained 24 ± 1℃. While in line 117, the authors stated that the average ambient temperature in the NC treatment was 24.59 ± 0.66℃.
Response 3: Thank you for pointing this out. In this study, the ambient temperature of NC treatment was maintained at 24±1℃, which is the design temperature of the experiment. The average ambient temperature of the NC treatment was 24.59±0.66℃, which was the actual measured temperature during the whole experiment period, as the temperature of the NC treatment was measured three times a day.
Comments 4: The description of results part was not accurate and should be improved. For example (line 235-236), the authors stated that the inclusion of tributyrin significantly enhanced FBW, ADG, and FCR of broilers during CHS (p < 0.05). The fact was that not all the level of tributyrin supplementation had the effect.
Response 4: Thank you for pointing this out. I agree with this comment. Therefore, I have modified. Line 237-239: supplementation with 0.16% and 0.32% tributyrin significantly enhanced FBW, ADG, and FCR of broilers during CHS (p < 0.05).
Line 244-245: supplementation with 0.08%, 0.16% and 0.32% of tributyrin increased CP digestibility in broilers under CHS (p < 0.05). (Line 246-248), supplementation with 0.16% and 0.32% of tributyrin increased serum HSP70 and Cort levels in broilers under CHS (p < 0.05).
Line 248-250: supplementation with 0.08%, 0.16% and 0.32% of tributyrin significantly reduced drip loss (p < 0.05) and increased the shear force value of breast muscle in broilers under CHS (p < 0.05).
Line 254-256: supplementation with 0.08%, 0.16% and 0.32% of tributyrin significantly reduced drip loss (p < 0.05) and increased the shear force value of breast muscle in broilers under CHS (p < 0.05).
Line 259-261: supplementation with 0.16% and 0.32% of tributyrin supplementation decreased EE content (p < 0.05) of breast muscle in broilers under CHS.
Line 265-267: supplementation with 0.16% and 0.32% of tributyrin decreased the MDA content and increased the GSH-Px activity of breast muscle in broilers under CHS (p < 0.05).
Line 271-273: supplementation with 0.16% and 0.32% of tributyrin up-regulated mRNA expressions of Nrf2, NQO1, HO-1, SOD, and GSH-Px (p < 0.05) of breast muscle in broilers under CHS.
Comments 5: Line 552. The authors stated that tributyrin supplementation, particularly at levels of 0.16% improved the meat quality by enhancing muscle antioxidant capacity. Why not 0.32%? How you come to this conclusion?
Response 5: Thank you for pointing this out. I agree with this comment. Therefore, I have modified the conclusion.
Line 555-561: In conclusion, dietary supplementation of tributyrin modulated gut microbiota imbalance, enhanced nutrient digestibility, and improved growth performance and meat quality in Taihe silky fowls under CHS. Furthermore, supplementation with 0.16% and 0.32% of tributyrin was effective in improving meat quality by enhancing muscle antioxidant capacity, which is believed to be associated with activating the Nrf2 signaling pathway. Considering the production cost comprehensively, the optimum supplemental level of tributyrin was 0.16% under the conditions of this study.

Reviewer 2 Report
Comments and Suggestions for Authors
General comment:
The heat stress is alarmingly approaching, and farmed animal experiments are justified to explore the mitigations possibilities. The present paper studies the sequential heat stress on local chicken breed with high economic importance (slow growth, high meat quality). The study analyses the heat stress on production traits, on microbiological composition of the gut and molecular mechanism of the effect and justifies the use of the different levels of trybutarate. The experiment is well designed and analysed correctly leading to a meaningful conclusion.
Specific comments, remarks:
The experimental housing needs to more detailed. How they can assure, or mad it possibel to have the same humidity level regardles of the temperature? Were the chicken kept in chambers?
Sample collection needs to be detailed. Were the individuals tagged, to ensure that the same bird was weighted before and after the experiment?
Why this starting age was chosen?
The feed intake was measured by cages, for the group of 30 birds. The average daily feed intake was the group feed intake divided by 28 days. The feed conversion rate was calculated by the average daily gain of the group divided by individual daily gain. This is not the individual feed conversion rate. The individual feed conversion rate is the individual feed intake divided by the individual growth.
The authors started to explain statistical testing in section 2.12, but the next section 2.13 is about statistical analysis. Better to move the last parapgraph to section 2.13 and have a clear description of the analysis. Differential analysis is used in economics and not in biological studies. The T-test is for comparing normal and heat stressed birds, analysis of variance is used for comparing the different levels of trybutirin.
line 420 stimu-lation instead of stim-ulation
line 439 „in” occours twice
line 459 little instead of litter
line 467 please rephrase „reported that revealed that”
line 526 thermo-neutral instead of thermos-neutral
The trybutirin showed its effect at/from 0.16% level. A paragraph in the discussion chapter could be devoted for the explanation. Why?
Author Response
Specific comments, remarks:
The experimental housing needs to more detailed. How they can assure, or mad it possibel to have the same humidity level regardles of the temperature? Were the chicken kept in chambers?
Response: Thank you for pointing this out. I agree with these comments. Therefore, I have added some information of the experiment design.
Line 109-114: The birds in the NC and HS treatments were housed in different facilities, and the temperature and relative humidity of the NC and HS treatments were measured three times a day. Continuous artificial light illuminated the interior space for the whole experiment period. The chicken houses were equipped with environmental control equipment, and the size of the cage is 70 (length) × 70 (width) × 40 (height) cm.
Sample collection needs to be detailed. Were the individuals tagged, to ensure that the same bird was weighted before and after the experiment?
Response: Thank you for pointing this out. I agree with these comments. Therefore, I have added some information of the sample collection.
Line 127-141: During the last 3 days of the experiment, the total excreta collection method was used for the determination of nutrient retention. During the test, feed intake and excrements were recorded daily, and the excreta were collected. Approximately 100 g of excreta and feed samples were collected. The samples were dried for 48 hours in a dryer at 65 °C, followed by 24 hours of equilibration to atmospheric conditions. Following this, the samples were pulverized using a Willey mill, passed through a 40-mesh screen, and stored at -20°C to analyze nutrient digestibility.
At the end of the experiment, after 12 hours fasting, 5 birds per treatment (one bird per cage was randomly selected and tagged) were collected blood samples (n = 30) by the wing vein. The serum was obtained through centrifugation (3500 r/min, 10 min, 4°C), and stored at -20°C for further analysis. After collecting blood samples, the broilers were slaughtered by severing the jugular vein, respectively. The meat quality of the slaughtered broilers was determined. Each broiler's entire left breast meat was promptly excised, and stored at 4 ℃ for subsequent analysis. Additionally, right breast muscle and cecal content samples were collected and rapidly stored in liquid nitrogen for further analysis.
Why this starting age was chosen?
Response: Thank you for pointing this out. It has been reported that heat stress impairs lipid metabolism in broilers by reducing protein deposition and promoting fat synthesis, thereby increasing fat percentage within the abdomen, subcutaneous, and muscle. At the same time, heat stress caused changes in blood parameters and some blood metabolites of broilers, indicating that blood metabolites are sensitive and valuable parameters for studying lipid metabolism in broilers exposed to heat stress. Guo et al. reported that heat stress for 4 weeks (28 days) led to the disorders of serum lipid metabolism in indigenous slow-growing broilers. In addition, the youth stage of Taihe silky fowls lasts from 12 to 20 weeks, and about 5-6 months is the market period of Taihe silky fowls. Therefore, we chose the experimental period (28 days) after 120 days from hatch.
Guo Y, Balasubramanian B, Zhao ZH, Liu WC. Heat stress alters serum lipid metabolism of Chinese indigenous broiler chickens-a lipidomics study. Environ Sci Pollut Res Int. 2021, 28(9):10707-10717. doi: 10.1007/s11356-020-11348-0.
The feed intake was measured by cages, for the group of 30 birds. The average daily feed intake was the group feed intake divided by 28 days. The feed conversion rate was calculated by the average daily gain of the group divided by individual daily gain. This is not the individual feed conversion rate. The individual feed conversion rate is the individual feed intake divided by the individual growth.
Response: Thank you for pointing this out. I agree with these comments. Therefore, I have modified. Line 143-145: The body weight (BW) was recorded on cage basis on day 1 and 28. Feed consumption was recorded on cage basis every week to calculate average daily feed intake (ADFI) and feed conversion ratio (FCR).
The authors started to explain statistical testing in section 2.12, but the next section 2.13 is about statistical analysis. Better to move the last parapgraph to section 2.13 and have a clear description of the analysis. Differential analysis is used in economics and not in biological studies. The T-test is for comparing normal and heat stressed birds, analysis of variance is used for comparing the different levels of trybutirin.
Response: Thank you for pointing this out. I agree with these comments. I have moved the section 2.12 last paragraph to section 2.13, and have redescribed the analysis, and line 127-141.
line 420 stimu-lation instead of stim-ulation
Response: Thank you for pointing this out. I have revised.
line 439 „in” occours twice
Response: Thank you for pointing this out. I have removed one “in”.
line 459 little instead of litter
Response: Thank you for pointing this out. I have revised.
line 467 please rephrase „reported that revealed that”
Response: Thank you for pointing this out. I have revised.
line 526 thermo-neutral instead of thermos-neutral
Response: Thank you for pointing this out. I have revised.
The trybutirin showed its effect at/from 0.16% level. A paragraph in the discussion chapter could be devoted for the explanation. Why?
Response: Thank you for pointing this out. I agree with this comment. The results of this study showed an effect of tributyrin at both 0.16% and 0.32% levels. Therefore, Therefore, I have modified the conclusion.
Line 555-561: In conclusion, dietary supplementation of tributyrin modulated gut microbiota imbalance, enhanced nutrient digestibility, and improved growth performance and meat quality in Taihe silky fowls under CHS. Furthermore, supplementation with 0.16% and 0.32% of tributyrin was effective in improving meat quality by enhancing muscle antioxidant capacity, which is believed to be associated with activating the Nrf2 signaling pathway. Considering the production cost comprehensively, the optimum supplemental level of tributyrin was 0.16% under the conditions of this study.

Reviewer 3 Report
Comments and Suggestions for Authors
The manuscript entitled " Dietary Tributyrin Improves Growth Performance, Meat Quality, Muscle Oxidative Status, and Gut Microbiota in Taihe Silky Fowls Under Cyclic Heat Stress" by Chuanbin Chen et al. The researcher addresses a topic of interest to Animal Journal readers and the poultry industry. The work tested the effects of adding tributyrin on growth performance, meat quality, muscle oxidative status, and gut microbiota of Taihe silky fowls under cyclic heat stress (CHS) conditions. Although I find this manuscript commendable, some items need revision /clarity.
Major
1. The author should give the reason for choosing the experimental period (28 days) after 120 days from hatch.
2. The author needs to justify picking the different levels of tributyrin (0.04%, 0.08%, 0.16%, and 0.32%), supported by past articles.
3. The discussion requires a more thorough review of relevant research with a stronger focus on Taihe silky fowl studies. Several arguments presented lack sufficient depth to support this study's results adequately.
Specific comments
L98, 101, Be consistent with using Taihe silky fowls or broilers throughout this report.
L294-361, It is not easy to follow the bars represented in Figures 2 to 8, I will suggest changing all the figures to tables for a better interpretation of the result.
L405, can you give the composition of the commercial diets given to the birds for the first 120 days before the experimental diets provided in Table 1?
L439, Remove one ‘in’
Line 467, rewrite the statement for clarity.
Line 416-548, Give more details relating to past work on Taihe silky fowls for in-depth discussion of the data.
Author Response
Major
- The author should give the reason for choosing the experimental period (28 days) after 120 days from hatch.
Response: Thank you for pointing this out. It has been reported that heat stress impairs lipid metabolism in broilers by reducing protein deposition and promoting fat synthesis, thereby increasing fat percentage within the abdomen, subcutaneous, and muscle. At the same time, heat stress caused changes in blood parameters and some blood metabolites of broilers, indicating that blood metabolites are sensitive and valuable parameters for studying lipid metabolism in broilers exposed to heat stress. Guo et al. reported that heat stress for 4 weeks (28 days) led to the disorders of serum lipid metabolism in indigenous slow-growing broilers. In addition, the youth stage of Taihe silky fowls lasts from 12 to 20 weeks, and about 5-6 months is the market period of Taihe silky fowls. Therefore, we chose the experimental period (28 days) after 120 days from hatch.
Guo Y, Balasubramanian B, Zhao ZH, Liu WC. Heat stress alters serum lipid metabolism of Chinese indigenous broiler chickens-a lipidomics study. Environ Sci Pollut Res Int. 2021, 28(9):10707-10717. doi: 10.1007/s11356-020-11348-0.
- The author needs to justify picking the different levels of tributyrin (0.04%, 0.08%, 0.16%, and 0.32%), supported by past articles.
Response: Thank you for pointing this out. I have added the relevant articles.
Line 105-106: The different levels of tributyrin addition refer to past studies [14, 20].
Gu, T.; Duan, M.; Liu, J.; Chen, L.; Tian, Y.; Xu, W.; Zeng, T.; Lu, L. Effects of tributyrin supplementation on liver fat deposition, lipid levels and lipid metabolism-related gene expression in broiler chickens. Genes, 2022, 13, 2219, doi: 10.3390/genes13122219.
Li, J.; Hou, Y.; Yi, D.; Zhang, J.; Wang, L.; Qiu, H.; Ding, B.; Gong, J. Effects of tributyrin on intestinal energy status, antioxi-dative capacity and immune response to lipopolysaccharide challenge in broilers. Asian-Australas J. Anim. Sci. 2015, 28, 1784-1793, doi: 10.5713/ajas.15.0286.
- The discussion requires a more thorough review of relevant research with a stronger focus on Taihe silky fowl studies. Several arguments presented lack sufficient depth to support this study's results adequately.
Response: Thank you for pointing this out. After searching the materials, I found that Taihe silky fowls are a unique breed in Jiangxi, China, and there are few international studies. I have added some studies relating to Taihe silky fowls.
Line 511-517: The meat of Taihe silky fowls contains melanin when compared with the other common chickens [6]. Natural melanin is considered as one of the most important components in Taihe silky fowls, which has a wide range of biochemical activities such as anti-oxidation and free radical-scavenging effects [5].
Liao, X.; Shi, X.; Hu, H.; Han, X.; Jiang, K.; Liu, Y.; Xiong, G. Comparative metabolomics analysis reveals the unique nutritional characteristics of breed and feed on muscles in Chinese Taihe black-bone silky fowl. Metabolites 2022, 12, 914, doi: 10.3390/metabo12100914.
Yuan, L.; Wu, H.; Wang, J.; Zhou, M.; Zhang, L.; Xiang, J.; Liao, Q.; Luo, L.; Qian, M.; Zhang, D. Pharmacokinetics, withdrawal time, and dietary risk assessment of enrofloxacin and its metabolite ciprofloxacin, and sulfachloropyridazine-trimethoprim in Taihe black-boned silky fowls. J. Food Sci. 2023, 88, 1743-1752, doi: 10.1111/1750-3841.16501.
Specific comments
L98, 101, Be consistent with using Taihe silky fowls or broilers throughout this report.
Response: Thank you for pointing this out. I have modified.
L294-361, It is not easy to follow the bars represented in Figures 2 to 8, I will suggest changing all the figures to tables for a better interpretation of the result.
Response: Thank you for pointing this out. I have changed the figures 2-7 to tables. By searching for relevant articles, I think Figure 8 can more intuitively reflect the differences of the result between treatment.
Lu, Z.; He, X.; Ma, B.; Zhang, L.; Li, J.; Jiang, Y.; Zhou, G.; Gao, F. Dietary taurine supplementation improves breast meat quality in chronic heat-stressed broilers via activating the Nrf2 pathway and protecting mitochondria from oxidative attack. J. Sci. Food. Agric. 2019, 99, 1066-1072, doi: 10.1002/jsfa.9273.
L405, can you give the composition of the commercial diets given to the birds for the first 120 days before the experimental diets provided in Table 1?
Response: Thank you for pointing this out. Sorry, due to commercial confidentiality, I cannot disclose the composition of the birds' commercial diets for the first 120 days.
L439, Remove one ‘in’
Response: Thank you for pointing this out. I have removed one “in”, and line 467.
Line 467, rewrite the statement for clarity.
Response: Thank you for pointing this out. I have rewritten the statement.
Line 463-465: Lactobacillus can assist hosts in obtaining nutritional support, reducing pH levels in the intestines, maintaining the mucosal barrier function, and protecting against damage from foreign substances and pathogens.
Line 416-548, Give more details relating to past work on Taihe silky fowls for in-depth discussion of the data.
Response: Thank you for pointing this out. I have added some studies relating to Taihe silky fowls in the discussion of the data.
Line 516-519: Liu et al. [4] also reported that a significant increase in MDA and lower levels of antioxidant enzymes in black-boned chickens exposed to high temperature. This may be due to the decrease of melanin synthesis in Taihe black-bone chicken due to heat stress.
Liu, L.L.; He, J.H.; Xie, H.B.; Yang, Y.S.; Li, J.C.; Zou, Y. Resveratrol induces antioxidant and heat shock protein mRNA ex-pression in response to heat stress in black-boned chickens. Poult. Sci. 2014, 93, 54-62, doi: 10.3382/ps.2013-03423.
